# Effect of A Rapid-Cooling Protocol on the Optical and Mechanical Properties of Dental Monolithic Zirconia Containing 3–5 mol% Y_2_O_3_

**DOI:** 10.3390/ma13081923

**Published:** 2020-04-19

**Authors:** Hee-Kyung Kim

**Affiliations:** Department of Prosthodontics, Institute of Oral Health Science, Ajou University School of Medicine, Suwon 16499, Korea; denthk@ajou.ac.kr; Tel.: +82-31-219-5322

**Keywords:** zirconium oxide, phase transition, optical phenomena, mechanical phenomena

## Abstract

Many attempts have been made to improve the translucency of zirconia in dentistry. The purpose of this study was to evaluate the effect of a rapid-cooling heat treatment on the optical and mechanical properties of dental monolithic zirconia. Zirconia containing 3, 4, and 5 mol% Y_2_O_3_ were sintered, sectioned, and polished. The specimens were rapidly cooled from high temperature inducing a diffusionless cubic-to-metastable tetragonal (t’) phase transformation. The changes in *L^*^a^*^b^*^* color coordinates, translucency parameter (TP), and total transmittance (T%) were measured. Three-point bending strength, Vickers hardness, and indentation fracture toughness tests were performed. Quantitative phase analyses were carried out by X-ray diffraction with Rietveld refinement. Scanning electron microscopy (SEM) images were obtained. With increasing Y_2_O_3_ contents, TP and T% values increased while strength and toughness decreased. The Rietveld analysis showed that the amount of t’-phase increased after rapid-cooling and annealed 5Y-partially stabilized zirconia (PSZ) contained the highest amount of t’-phase (64.4 wt%). Rapid-cooling improved translucency but the translucency of annealed 5Y-PSZ did not approach that of lithium disilicate glass-ceramic. Rapid-cooling decreased flexural strength significantly, being 306.1 ± 61.8 MPa for annealed 5Y-PSZ. SEM revealed that grains tended to get larger after rapid-cooling. A rapid-cooling treatment can produce t’-phase which can contribute to an increase in translucency but has a negative effect on the mechanical properties of zirconia.

## 1. Introduction

Over the past 10 years, the use of yttria-stabilized zirconia (YSZ) in a monolithic design has been widely considered in dental restorations. The major properties that motivates this choice are the conservative tooth reduction, minimal wear on the opposing dentition, and the reduced risk for veneer chipping. However, the lack of translucency of zirconia due to the light scattering at the grain boundaries, residual pores, and secondary phases is an inherent disadvantage [1,2] and thus several attempts, such as a reduced Al_2_O_3_ content, an addition of La_2_O_3_ into 3 mol% yttria-stabilized tetragonal zirconia polycrystal (3Y-TZP), or a higher yttria (Y_2_O_3_) content have been made [3,4]. 

Along with those attempts, the changes in the mechanical and aging behaviors have been addressed. The reduced amounts of Al_2_O_3_ to less than 0.05% could contribute to increasing translucency while mechanical properties slightly decreased [5,6]. By incorporating La_2_O into 3Y-TZP, the segregation of Al_3_^+^ and La_3_^+^ to the grain boundary could limit oxygen vacancy annihilation resulting in improved aging resistance [3]. Furthermore, La_2_O doping into 3Y-TZP improved translucency, but also induced the deterioration of mechanical properties [3]. However, those investigated approaches using certain dopants (Al_2_O_3_ or La_2_O) appeared to have insufficient translucency for anterior dental restorations. 

The markedly improved translucency was achieved by using a higher yttria content. Pure zirconia changes its crystal structures depending on the temperature: monoclinic (m), tetragonal (t), and cubic (c). Additions of some oxides (MgO, CaO, or Y_2_O_3_) to pure zirconia can limit phase transitions and lead to stabilize either cubic or tetragonal form at room temperature. With 3 mol% yttria, a metastable tetragonal single phase could be retained. By increasing yttria contents up to 5 mol%, Y_2_O_3_-ZrO_2_ were heat treated at equilibrium temperature of a two-phase mixture (cubic + tetragonal) and then on cooling, cubic phase was partially stabilized. Accordingly, partially stabilized zirconia (PSZ) with a mixed-phase (c + t) can be prepared [7]. The cubic structure of zirconia is optically isotropic, so that light scattering at grain boundaries is less critical. As a result, translucency can be vastly improved with the new cubic phase-containing zirconia, which is comparable to lithium disilicate [6,8]. Since the high toughness of PSZ system is attributed to the stress-induced martensitic transformation of the tetragonal phase, cubic phase which is resistant to phase transformations under stresses can have a detrimental effect on the mechanical properties [9]. Therefore, a decrease in the mechanical properties of high-translucent zirconia can be an important concern.

In the Y_2_O_3_-ZrO_2_ system, there are two types of phases: the tetragonal phase (t) which can be formed by controlled diffusion and the metastable tetragonal phase (t’) which can be formed by a diffusionless transition from the cubic phase or in a mixed-phase (c + t) [10]. The t’-phase can be produced by rapid-cooling of compositions in the range of 3–7 mol% yttria at temperature above 1425 °C [11]. In terms of the t → m transformation behavior, the t’-phase is referred to as non-transformable due to its resistance to transformation under stress [10,12] and hence, it can be more stable to hydrothermal aging [13]. Although the t’-phase does not exhibit stress-induced transformation toughening [10], it has another toughening mechanism, i.e., ferroelastic domain switching, by domain reorientation without a change of crystal structure in compression [14]. Thus, the changes of domain orientation under external stresses could contribute to toughness by releasing the lattice strain [15].

The t’-phase has a higher yttria content than t-phase, resulting in a smaller tetragonality (*c/a* ratio) which is close to unity [11]. Therefore, it can be hard to distinguish between t’- and c-phases by using X-ray diffraction (XRD) and these two phases may be discriminate by comparing the *c/a* ratios [10]. Due to its smaller tetragonality in terms of a lattice parameter, the t’-phase might cause less light scattering than t-phase allowing more light transmission through zirconia. Taking advantage of the temperature-dependent phase transformation behavior of zirconia, the author induced t’-phase and investigated the role of t’-phase in dental zirconia in this report. The purposes of this in vitro study were therefore to analyze the crystalline phases of dental monolithic zirconia with 3–5 mol% Y_2_O_3_ and to verify the effect of a rapid-cooling heat treatment on the crystalline structures, optical, and mechanical properties of dental monolithic zirconia. The null hypothesis to be tested was that the rapid-cooling protocol would not affect the crystalline structures, optical, and mechanical properties of dental monolithic zirconia containing 3–5 mol% Y_2_O_3_.

## 2. Materials and Methods

### 2.1. Specimen Preparation

Samples of polycrystalline zirconia containing 3, 4, and 5 mol% Y_2_O_3_ (Luxen Zr, Luxen Enamel, and Luxen Smile, respectively; DENTALMAX, Seoul, Korea) were sintered in air at 1500 °C for 2 h and subsequently air cooled. The sintered blocks were then cut into plates and bars to allow the analyses. Each surface of the sectioned specimen was polished to a mirror finish (1 μm) with a lapping and polishing machine (SPL-15 GRIND-X; OKAMOTO, Japan). The final sizes of the thin plates were 17 × 17 × 1 mm (n = 10 for each yttria group) and those of the thick plates were 17 × 17 × 5 mm (n = 10 for each yttria group). The final dimension of the bar specimens was 4 × 3 × 45 mm (n = 20 for each yttria group). 

Lithium disilicate glass-ceramic (IPS e.max CAD; Ivoclar Vivadent AG, Schaan, Liechtenstein) were also prepared and served as control; 14 × 14 × 1 mm (n = 10), 4 × 3 × 45 mm (n = 10), and 14 × 14 × 5 mm (n = 5). The specimens underwent a crystallization process in a furnace (Programat P310; Ivoclar Vivadent AG, Schaan, Liechtenstein) according to the manufacturer’s instructions. The materials used in the present study and their sintering conditions are listed in Table 1. The experimental study design is presented in Figure 1.

### 2.2. Annealing at 1550 °C Followed by Rapid-Cooling

The zirconia specimens were heated in air for 1 h at 1550 °C in order to being employed inside the two-phase (c + t) field, taking into account Scott’s phase diagram (Figure 2) [7]. Then they were rapidly air-cooled within 1–2 min by taking them out from the furnace and dropping them onto a brass plate to prevent diffusional phase separation [11,14,16,17].

### 2.3. Optical Properties Determination

For sintered/polished thin plate specimens, spectral reflectance data in the wavelength range of 360–750 nm at 10-nm intervals were acquired with a spectrophotometer (Ci7600; X-Rite, Grand Rapids, MI, USA) against a white polytetrafluoroethylene background (GM29010330; X-Rite, Grand Rapids, MI, USA; CIE *L^*^* = 94.26, *a^*^* = −0.24, *b^*^* = 0.95), a black glass-ceramic tile (CM-A101B; Konica Minolta Optics Inc., Tokyo, Japan; CIE *L^*^* = 0.11, *a^*^* = −0.08, *b^*^* = 0.02), and an A2 glass-ceramic tile (IPS e.max Press MO; Ivoclar Vivadent AG, Schaan, Liechtenstein; CIE *L^*^* = 76.32, *a^*^* = −2.00, *b^*^* = 9.88).

CIELab color coordinates relative to D65 with the 2-degree standard observer (CIE 1931) and diffuse/8-degree geometry were calculated from the reflectance data. A 6-mm diameter aperture and a 6-mm diameter measurement area were used. The measurements were conducted by using glycerin as an optical coupling medium in-between the specimen and the background [18]. The translucency parameters (TP) were obtained by calculating the CIEDE2000 color differences ($∆E_{00}$) between the values against white and black backgrounds [19], as is defined as follows:(1)$$∆E_{00}={\left[{\left(\;\frac{∆L^{\prime }}{K_{L}S_{L}}\right)}^{2}+{\left(\;\frac{∆C^{\prime }}{K_{C}S_{C}}\right)}^{2}+{\left(\;\frac{∆H^{\prime }}{K_{H}S_{H}}\right)}^{2}+R_{T}\left(\;\frac{∆C^{\prime }}{K_{C}S_{C}}\right)\left(\;\frac{∆H^{\prime }}{K_{H}S_{H}}\right)\right]}^{\frac{1}{2}}$$
where ${\Delta C}^{\prime }$ and $\Delta H^{\prime }$ are the differences in chroma and hue for a pair of samples. $S_{L}$, $S_{C}$, and $S_{H}$ are the weighting functions for the lightness, chroma, and hue and the parametric factors, $K_{L}$, $K_{C}$, and $K_{H}$ are the correction terms for variations in experimental conditions. $R_{T}$, a rotation function, accounts for the interaction between chroma and hue differences in the blue region.

The total transmittance (T%) was calculated using the following formula [20]:(2)$$T\%=(\frac{L^{*}{}_{\mathrm{specimen}}}{L^{*}{}_{\mathrm{source}}})\times 100$$

The *L*^*^_source_ was obtained without any specimen placed before each measurement. The *L*^*^_specimen_ was recorded from 360 to 750 nm and the mean T% values at 550 nm wavelength were used to compare the test specimens [21,22]. All measurements were repeated three times on each specimen.

After rapid-cooling heat treatments, spectral reflectance data, TP, and T% values were obtained for each specimen group according to the same protocol as those used for sintered/polished specimens.

### 2.4. Mechanical Properties Measurement

The hardness and fracture toughness were measured by Indentation Fracture (IF) method on the thick plate specimen using a Vickers microhardness tester (Z2.5; Zwick/Roell, Ulm, Germany) with a load of 10 kg for 10 s. Five indentations were used to determine the average hardness. Flexural strengths of the bar specimens were measured by using a 3-point bending test with a 30-mm span according to International Organization for Standardization standard 6872:2015 [23]. The specimens were loaded in a universal testing machine (RB 301 UNITECH-M; R&B, Daejeon, Korea) with a 5-kN load cell (UM-K500; DACELL, Cheongju, Korea) at a crosshead speed of 0.5 mm/min until fracture occurred. The bending tests were performed under controlled temperature (22 ± 1 °C) and humidity (31 ± 1%).

### 2.5. Crystalline Phase Analysis

To identify the crystalline phase of zirconia specimens, one specimen from each zirconia group (sintered/polished or rapidly-cooled) was randomly selected. XRD (D8 Advance; Bruker AXS, Karlsruhe, Germany) was conducted by using Cu-Kα radiation (λ = 1.54056Å) at 40 kV and 40 mA. XRD profiles were collected between 25 and 80° (2θ) at a scan speed of 1°/min in a continuous-scanning mode with a step size of 0.01°.

The relative phase contents of monoclinic (m), tetragonal (t), and cubic zirconia (c) were determined by Rietveld refinements with Topas academic software (Bruker AXS, Karlsruhe, Germany). The phase structures were refined as: tetragonal zirconia unit cell with space group P4_2_/nmc, monoclinic zirconia unit cell with space group P2_1_/c, cubic zirconia unit cell with space group Fm$\bar{3}$m. The quality of the Rietveld refinement was controlled with a R value to be less than 10%.

### 2.6. Microstructural Analysis

The microstructures of the thermally-etched (zirconia) or acid-etched (lithium disilicate) specimens were observed using a scanning electron microscope (SEM; JSM-7900F; JEOL Ltd., Tokyo, Japan). For zirconia, thermal etching was carried out at 1350 °C for 30 min with fast heating rate (20 °C/min) to prevent grain growth. After coating with a thin layer of platinum (108 auto sputter coater; Cressington, Watford, UK), secondary electron SEM images were acquired in vacuum (9.6 × 10^−5^ Pa) with an acceleration voltage of 15.0 kV and an emission current of 64.2 µA.

### 2.7. Statistical Analysis

Statistical analyses were conducted with statistical software (IBM SPSS Statistics, v25.0; IBM Corp., New York, NY, USA). The Kolmogorov-Smirnov test was done to determine the normality assumption. A two-way ANOVA was performed to assess the interrelationship of two independent variables (yttria content and rapid-cooling heat treatment) on the optical or mechanical properties of zirconia. The significance level was set at 0.05.

## 3. Results

### 3.1. Optical properties

Means of CIE *L^*^*, *a^*^*, and *b^*^* against an A2 background, TP, and T% at 550 nm of each group are reported in Table 2. A two-way ANOVA test revealed that there were statistically significant interactions between yttria content and a rapid-cooling heat treatment on *a^*^*; *F* (2, 174) = 170.412, *p* < 0.001, partial η^2^ = 0.662, observed power = 1.000 and *b^*^*; *F* (2, 174) = 58.798, *p* < 0.001, partial η^2^ = 0.403, observed power = 1.000. Simple main effects analyses showed that there were significant differences in *a^*^* values between sintered/polished and rapid-cooling groups for 3Y- and 4Y-PSZ (*p* < 0.001) and there were significant differences in *a^*^* values among 3Y-, 4Y-, and 5Y- groups for sintered/polished (*p* < 0.001) and rapid cooling groups (0.001 < *p* < 0.006). Simple main effects analyses showed that there were no significant differences in *b^*^* values among 3Y-, 4Y-, and 5Y- groups for rapid cooling groups (0.157 < *p* < 0.798). There were no significant interactions between the effect of yttria content and a rapid-cooling heat treatment on *L^*^*; *F* (2, 174) = 0.094, *p* = 0.911, partial η^2^ = 0.001, observed power = 0.064, TP; *F* (2, 174) = 0.164, *p* = 0.849, partial η^2^ = 0.002, observed power = 0.075, and T%; *F* (2, 174) = 2.288, *p* = 0.104, partial η^2^ = 0.026, observed power = 0.460. According to Tukey post hoc test, *L^*^* value decreased, TP value increased, and T% increased as the yttria contents increased for sintered/polished. The additional rapid cooling treatment for each zirconia group improved translucency, as indicated in Figure 3d,e,h. The changes in optical properties based on the results of a two-way ANOVA test are shown in Figure 3.

### 3.2. Mechanical Properties

The mechanical properties of each ceramic group are reported in Table 3. Two-way ANOVA analyses indicated significant interactions between yttria content and a rapid-cooling heat treatment on the flexural strength and the fracture toughness (*p*
≤ 0.001). For sintered/polished zirconia specimens, flexural strength and fracture toughness decreased as the yttria contents increased. The flexural strength was significantly reduced after a rapid-cooling heat treatment, while the fracture toughness was significantly increased after rapid-cooling for all zirconia grades. 3Y-TZP had the highest flexural strength, but the strength sharply decreased down to 14.7%. The hardness remained relatively stable during rapid-cooling.

### 3.3. Phase Characteristics 

Figure 4 exhibits diffraction patterns of each zirconia group obtained over the 2θ range 25°-80°. No monoclinic phase was identified and only diffraction peaks corresponding to the low-yttria t-phase and the high-yttria t’-phase were detected. From the results in Figure 4, the higher yttria sample exhibited a greater amount of t’-phase. For the samples with rapid-cooling heat treatments, the intensities of the diffraction peaks corresponding to the t’-phase increased with simultaneous decreases in those corresponding to the t-phase.

The quantitative phase analysis by using the Rietveld method indicated that t’-phases were present in all samples. 3Y-TZP contained 79.2 wt% t-phase and 20.8 wt% t’-phase, and the t’-phase increased to 59.1 wt% for 5Y-PSZ. As the yttria contents increased, the *c/a* ratio (tetragonality) of the t’-phase decreased, approaching unity. Rapid-cooling of the sample caused an increase in the t’-phase contents and R/5Y-PSZ contained the highest amount of t’-phase (64.4 wt%). The results are summarized in Table 4.

### 3.4. Microstructure

SEM images of each ceramic group were shown in Figure 5. The grain sizes of sintered/polished 3Y-TZP ranged from about 280 to 350 nm, whereas larger grains were observed in 4Y- (416–890 nm) or 5Y-PSZ (543–1060 nm). There was also a tendency to form larger grains after a rapid-cooling heat treatment. The SEM image demonstrated that R/5Y-PSZ had the largest grains (0.88–1.60 µm).

## 4. Discussion

In order to further improve the translucency of dental monolithic zirconia, the author fabricated t’-phase-containing translucent zirconia. According to the results of this study, a rapid-cooling protocol found to be detrimental to the mechanical properties even though it improved the translucency of dental monolithic zirconia containing 3–5 mol% Y_2_O_3_. Therefore, the null hypothesis was rejected.

Jue et al. [14] reported that above 1400 °C, re-crystallization of Y_2_O_3_-ZrO_2_ samples caused the formation of new domains with different texture (t’-phase). The t’-phase was induced by the displacement of oxygen ions in the lattice parameter through a diffusionless mechanism. The presence of t’-phase has been verified by its characteristic platelike domain structure with antiphase boundaries [16]. Due to the strong overlapping diffraction patterns of the c-phase and t’-phase [11], it was difficult to distinguish between them. In this study, zirconia samples were subjected to rapid-cooling heat treatments in an attempt to form t’-ZrO_2_ and the weight fractions of each phase were quantified using the Rietveld refinement method. Zhang and Lawn [5] reported that with a higher yttria content (4 or 5 mol%), cubic phase could be stabilized yielding high-translucent partially-stabilized zirconia. Unlikely, the results of this study showed that sintered (cooled at a cooling rate of 8°/min)/polished Y_2_O_3_-ZrO_2_ samples had two types of tetragonal phases (t + t’) and no monoclinic or cubic phases were identified. With increasing yttria contents (from 3–5 mol%), the proportion of t’-phase increased at the expense of t-phase. The t’-phase is morphologically different from t-phase even though crystallographically identical. In addition, *c/a* ratio (tetragonal axial factor) of t’-phase approached unity with increasing yttria contents. Due to the close proximity of its crystal parameter to the isotropic structure, light scattering from birefringence can be reduced resulting in enhanced translucency of dental zirconia. In this study, annealing at 1550 °C allowed the material to reach its equilibrium state and upon rapid cooling, no t → m transformation occurred. From the c + t two-phase region, c-phase went through a diffusionless c → t’ transition and the t-phase stayed up to room temperature. By a rapid-cooling heat treatment protocol, the t’-ZrO_2_ was formed for all zirconia groups and thus, the enhanced translucency was obtained.

Since cubic zirconia do not undergo stress-induced transformation [3], it adversely affects the mechanical properties. The t’-phase is quite resistant to t → m phase transformation, while it can undergo ferroelastic domain switching strengthening mechanism under stress due to ferroelastic nature of tetragonal zirconia [14]. In this study, fracture toughness values increased after a rapid-cooling heat treatment. For 5Y-PSZ, fracture toughness increased more than twice assuming the contribution of domain re-orientation to toughness. However, flexural strength significantly decreased especially for 3Y-TZP. Due to the sudden temperature change during cooling, residual stresses might be induced leading to subcritical crack propagations. As seen in SEM images in this study, zirconia with a higher yttria content had larger grains and the grains tended to get larger after rapid-cooling. However, those larger grains had a detrimental effect on the strength of zirconia and the result was consistent with the findings of the previous study [4], although the increased grain size minimized grain-boundary light scattering yielding more light transmission.

Based on the results of this study, increased yttria contents reduced the lightness color value of zirconia. In addition, a rapid-cooling heat treatment also affected color values of zirconia. The lightness increased, while the chroma decreased. The translucency was improved by a rapid-cooling heat treatment although the value did not reach that of lithium disilicate. Unlike the t-phase, the stable t’-phase does not suffer from long-term thermal aging which causes a degradation of the mechanical properties [13].

The application of the t’-phase would be one of the strategies to increase the translucency of Y_2_O_3_-ZrO_2_. However, according to the results of this study, a rapid temperature drop in the cooling process weakened the mechanical properties. Therefore, more effective cooling protocols to yield t’-phase without sacrificing high toughness should be investigated. After heating at the temperature inside the two-phase field (following Scott [7]), cooling immediately to 1400 °C to avoid the precipitation of t-phase through a diffusive transition, followed by subsequent furnace cooling to room temperature might reduce residual stresses. Further studies should be carried out.

The present study has some limitations. Indentation Fracture (IF) method was used to test fracture toughness. A recent study [24] reported that IF method did not provide accurate values and Single Edge Precracked Beam (SEPB) method could be a good option for Y-TZP even though it was difficult to produce precracks. In addition, the small-window color measurements could cause an edge loss of the light, resulting in the loss of spectral information.

Although there have been much research on the fabrication of high-translucent monolithic zirconia with an addition of dopant element, there have been no research on the production of t’-phase-containing translucent zirconia. Due to the high overlapping diffraction pattern with c-phase reflection, t’-phase can be easily missed. In this study, it was found that sintered/polished dental monolithic zirconia with 3–5 mol% Y_2_O_3_ contained t’-phase instead of c-phase, as observed by the Rietveld analysis. Particularly, the author tried to produce t’-phase through a rapid cooling protocol according to the temperature-dependent phase transition mechanism of zirconia. Based on the results obtained, the formation of t’-phase increased the translucency of zirconia and t’-phase-containing translucent monolithic zirconia could be fabricated not with an addition of dopant element but with the change of phase composition through a specific cooling protocol.

## 5. Conclusions

In summary, the author fabricated t’-phase-containing translucent monolithic zirconia by altering phase compositions using a rapid-cooling protocol. The t’-phase can improve the translucency, while its mechanical properties were sacrificed. The optimal protocol to improve the translucency without deteriorating its mechanical characteristics of dental monolithic zirconia will be determined in the future.

## Figures and Tables

**Figure 1 materials-13-01923-f001:**
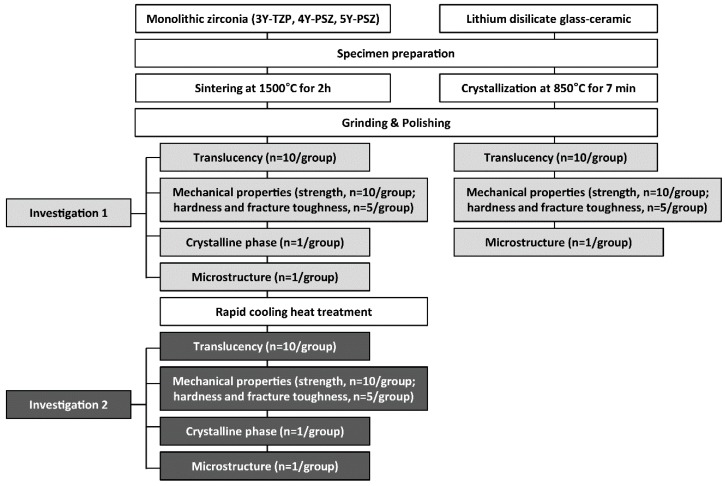
The experimental study design.

**Figure 2 materials-13-01923-f002:**
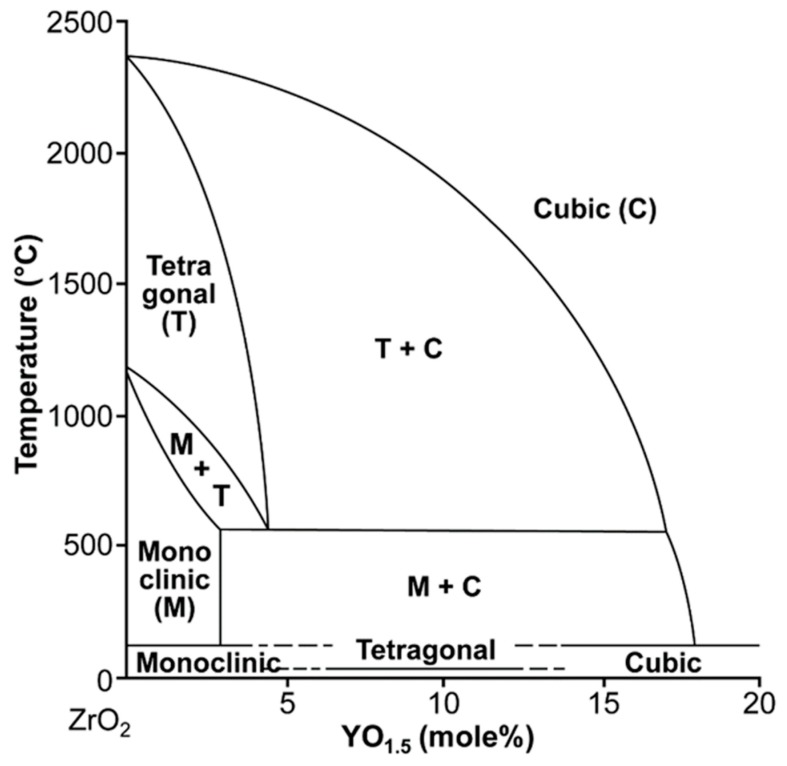
Zirconia-yttria phase diagram presented by Scott in 1975 [7].

**Figure 3 materials-13-01923-f003:**
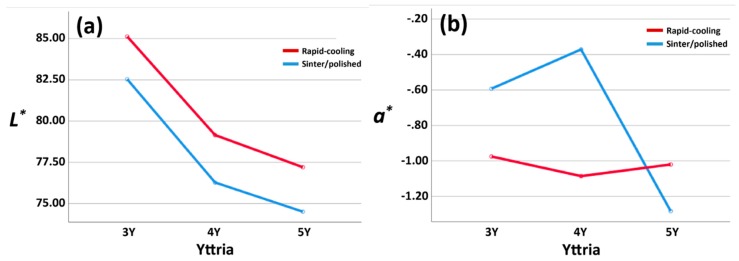
The changes of *L^*^* (**a**), *a^*^* (**b**), *b^*^* (**c**), TP (**d**), T%550 (**e**) values based on the results of a two-way ANOVA test. The changes in color (**f**,**g**) and total transmittance (**h**) after a rapid-cooling heat treatment.

**Figure 4 materials-13-01923-f004:**
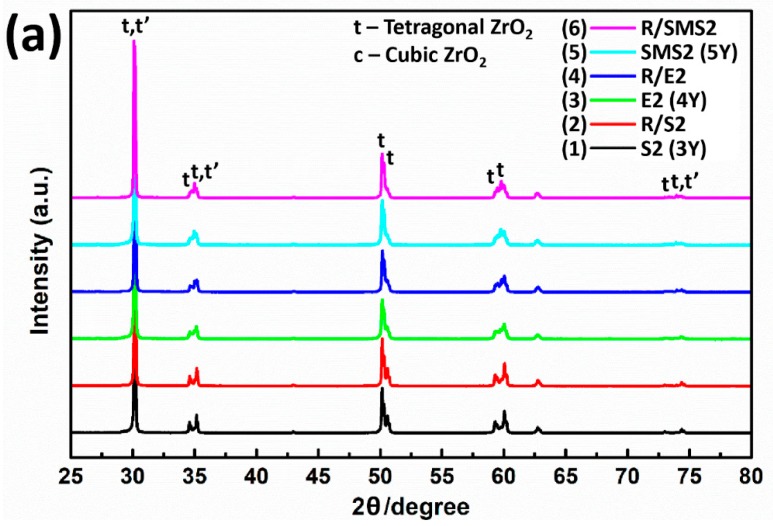
(**a**) The 25°–80° 2θ range of XRD patterns of each zirconia group. Expanded view of the (**b**) 29.8–30.6, (**c**) 34.4–35.6, and (**d**) 72.0–75.0 2θ range.

**Figure 5 materials-13-01923-f005:**
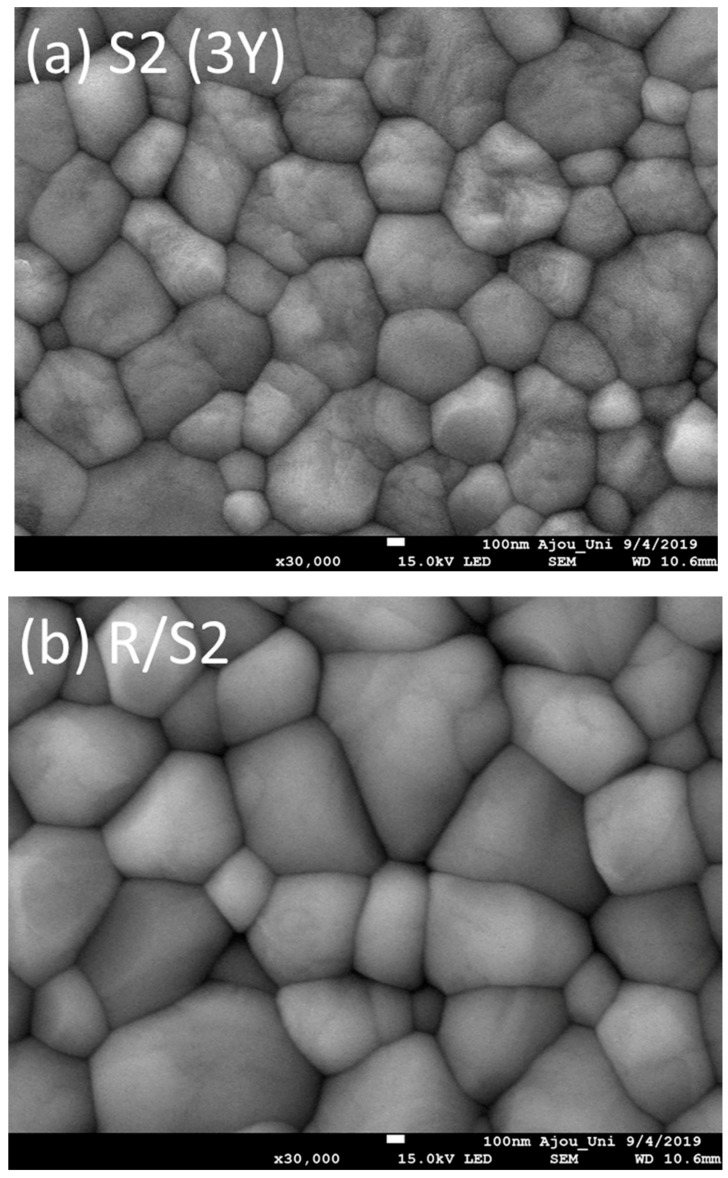
Scanning electron micrographs of the ceramic specimens investigated: (**a**) S2 (3Y), (**b**) R/S2, (**c**) E2 (4Y), (**d**) R/E2, (**e**) SMS2 (5Y), (**f**) R/SMS2, and (**g**) e.max.

**Table 1 materials-13-01923-t001:** Properties of materials used in this study.

Materials	Manufacturer	Shade	Batch No.	Composition (wt%)	Sintering Condition
Zirconia					
Luxen Zr (3Y-TZP)	DENTALMAX	S2	160523-S2-2	Y_2_O_3_: 5.35, Al_2_O_3_: 0.05, SiO_2_, Fe_2_O_3_ ≤ 0.002	1500 °C for 2 h
Luxen Enamel (4Y-PSZ)	DENTALMAX	E2	190327-10E2P-6	Y_2_O_3_: 6.95, Al_2_O_3_: 0.05, SiO_2_, Fe_2_O_3_ ≤ 0.002	1500 °C for 2 h
Luxen Smile (5Y-PSZ)	DENTALMAX	SMS2	190222-10SMS2-1	Y_2_O_3_: 9.32, Al_2_O_3_: 0.05, SiO_2_, Fe_2_O_3_ ≤ 0.002	1500 °C for 2 h
Glass-Ceramic					
IPS e.max CAD	Ivoclar Vivadent	HT A2	T02466 V35858 X17428	SiO_2_: 57.0–80.0, Li_2_O: 11.0–19.0, Other oxides	820 °C for 2 min + 840 °C for 7 min

**Table 2 materials-13-01923-t002:** Means (standard deviation) of CIE *L^*^*, *a^*^*, and *b^*^* against an A2 background, TP, and T% at 550 nm of each group.

Materials		*L^*^*	*a^*^*	*b^*^*	TP	T%
**Zirconia**						
3Y-TZP	S2	82.54 (1.62)	−0.59 (0.22)	11.11 (1.81) ^a^	4.43 (1.62) ^a^	30.86 (5.82)
	R/S2	85.13 (1.59)	−0.98 (0.11)	7.49 (0.63) ^b^	4.65 (0.39) ^a^	32.59 (4.73)
4Y-PSZ	E2	76.28 (1.88)	−0.37 (0.04)	13.77 (0.97)	8.47 (1.38)	44.13 (5.27)
	R/E2	79.15 (0.80)	−1.09 (0.13) ^a^	7.97 (0.50) ^b^	8.93 (0.43)	49.39 (2.99)
5Y-PSZ	SMS2	74.51 (2.02)	−1.28 (0.09)	11.50 (0.66) ^a^	9.37 (1.31)	51.08 (4.38)
	R/SMS2	77.21 (2.51)	−1.03 (0.16) ^a^	9.47 (0.45)	9.66 (1.06)	53.94 (1.06)
**Glass-Ceramic**						
e.max CAD	HT A2	65.81 (0.51)	−1.13 (0.04)	8.64 (0.27)	17.42 (0.29)	86.17 (0.91)

R/S2, R/E2, R/SMS2, rapidly-cooled S2, E2, SMS2, respectively. Means with the same lower-case superscript letter in each column are not significantly different from each other (*p* > 0.05).

**Table 3 materials-13-01923-t003:** Physical and mechanical properties of the specimens investigated. Standard deviations in parentheses.

Materials		Density (g/cm^3^)	Modulus (GPa)	Hardness (GPa)	Strength (MPa)	Toughness (MPa m^1/2^)
**Zirconia**						
3Y-TZP	S2	6.096	208	12.74 (0.05)	1054.4 (68.1)	4.34 (0.09)
	R/S2	6.077	211	12.4 (0.19)	154.7 (84.1)	5.74 (0.17)
4Y-PSZ	E2	6.108	212	13.08 (0.13)	1038.4 (55.4)	3.54 (0.13)
	R/E2	6.087	211	12.78 (0.04)	256.3 (77.1)	4.80 (0.20)
5Y-PSZ	SMS2	6.100	214	13.16 (0.15)	801.7 (64.5)	3.18 (0.13)
	R/SMS2	6.057	311	12.82 (0.08)	306.1 (61.8)	6.88 (0.38)
**Glass-Ceramic**						
e.max CAD	HT A2	2.502	102	5.72 (0.08)	288.5 (31.0)	2.34 (0.32)

**Table 4 materials-13-01923-t004:** Phase compositions and lattice parameters obtained by Rietveld analysis.

Parameter	3Y-TZP		4Y-PSZ		5Y-PSZ	
	S2	R/S2	E2	R/E2	SMS2	R/SMS2
*R_wp_* (%)	5.77	6.00	5.60	5.88	5.62	5.90
*R_exp_* (%)	2.71	1.52	2.73	1.51	2.65	1.51
*R_p_* (%)	4.51	4.69	4.41	4.42	4.27	4.56
GOF	2.13	3.95	2.05	3.89	2.12	3.90
**t-phase**						
z (O)	0.5387(7)	0.5418(10)	0.5368(11)	0.5457(14)	0.5472(12)	0.5485(18)
a (Å)	3.6045(2)	3.60489(3)	3.6061(4)	3.60599(5)	3.6058(3)	3.60501(5)
c (Å)	5.1787(4)	5.17902(6)	5.1778(7)	5.17898(10)	5.1778(7)	5.17832(10)
Amount (wt%)	79.2(5)	73.9(5)	60.0(6)	48.4(5)	40.9(3)	35.6(5)
*c/a* ratio	1.0159	1.0159	1.0153	1.0156	1.0154	1.0157
**t’-phase**						
z (O)	0.466(3)	0.478(6)	0.472(2)	0.485(4)	0.482(2)	0.479(2)
a (Å)	3.6218(11)	3.62105(10)	3.6224(7)	3.62192(6)	3.6238(4)	3.62219(5)
c (Å)	5.1533(2)	5.1548(2)	5.1524(14)	5.15502(13)	5.1527(8)	5.15506(10)
Amount (wt%)	20.8(5)	26.1(5)	40.0(6)	51.6(5)	59.1(3)	64.4(5)
*c/a* ratio	1.0061	1.0066	1.0058	1.0064	1.0054	1.0063

Space group = P4_2_/nmcz (origin choice 2); Values in parentheses correspond to the estimated standard deviation in the least significant figure to the left. *c/a* ratio = c (Å)/$\sqrt{2}$ a (Å); GOF = goodness of fit.

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
