# Peer review of "Effect of A Rapid-Cooling Protocol on the Optical and Mechanical Properties of Dental Monolithic Zirconia Containing 3–5 mol% Y2O3"

_materials, 2020, doi:10.3390/ma13081923_

Round 1

Reviewer 1 Report

The author demonstrated the optical and mechanical properties of zirconia containing 3 to 5 mol% Y2O3 after rapid-cooling. Although the methods and results are clearly stated, the rapid-cooling effect on the mechanical and optical properties were unclear. In my opinion, a substantial revision is needed to make this manuscript suitable for publication.

  1. Why did the author heat the samples up to 1550ºC before rapid-cooling? Higher sintering temperature induce crystal growth and changes the translucency.
  2. The author should clearly state the effect of rapid-cooling effect on optical properties in Results part.
  3. Please show the statistical refinement parameters not only Rwp but also Re and goodness-of-fit.
  4. It seems to me that the rapid-cooling has only negative effect on the dental prosthesis made with dental zirconia. What was the aims of this study?
  5. The resolution of Figure 1, 2, and 4 is low. Please change with high-resolution data.
  6. Please rearrange the figure numbers in Figure 5.

Author Response

I, the author, highly appreciate the detailed valuable comments on this manuscript.

The suggestions are quite helpful for me and I incorporate them in the revised paper.

The revision was listed below the comments and recommendations one by one.

================================================================Why 1. Why did the author heat the samples up to 1550ºC before rapid-cooling? Higher sintering temperature induce crystal growth and changes the translucency.

I followed the sintering schedule according to the manufacturer’s instruction. In an attempt to improve monolithic zirconia with acceptable translucency, processing of the new generation of 3Y-TZPs was refined largely by eliminating porosity by sintering at a higher temperature [Zhang, Y.; Lawn, B.R. Novel zirconia materials in dentistry. J. Dent. Res. 2018, 97, 140-147.]

2.The author should clearly state the effect of rapid-cooling effect on optical properties in Results part.

We described the effect of rapid-cooling effect on optical properties in Results part of the revised manuscript: The additional rapid cooling treatment for each zirconia group improved translucency, as indicated in Figure 3d, 3e, and 3h.

 3. Please show the statistical refinement parameters not only Rwp but also Re and goodness-of-fit.

Rexp

Rwp

Rp

GOF

S2

2.71

5.77

4.51

2.13

R/S2

1.52

6.00

4.69

3.95

E2

2.73

5.60

4.41

2.05

R/E2

1.51

5.88

4.42

3.89

SMS2

2.65

5.62

4.27

2.12

R/SMS2

1.51

5.90

4.56

3.90

In the manuscript, Rwp was mentioned as a representative parameter in Table 4.

4.It seems to me that the rapid-cooling has only negative effect on the dental prosthesis made with dental zirconia. What was the aims of this study?

In order to further improve the translucency of dental monolithic zirconia, the author fabricated t’-phase-containing translucent zirconia. Based on the results obtained, the formation of t’-phase increased the translucency of zirconia and t’-phase-containing translucent monolithic zirconia could be fabricated not with an addition of dopant element but with the change of phase composition through a specific cooling protocol.

5. The resolution of Figure 1, 2, and 4 is low. Please change with high-resolution data.

I am sorry. I changed Figure 1, 2, and 4 with high-resolution data already and no more high resolution could be achieved.

6.,Please rearrange the figure numbers in Figure 5.

Yes, I did.

Reviewer 2 Report

This is an interesting work on the effects of inserting different percentages of Y2O3 on the optical and mechanical properties of zirconia for dental use.

Some criticisms are present:

-The author's affiliation is missing

-In the abstract a general sentence on the problem should be inserted

-In the abstract section remove each acronym

-Introduction: some general considerations on the advantages and disadvantages of zirconia compared, for example, to lithium disilicate, should be provided.

Especially considering CAD CAM technologies which best choice for the clinician? And what vatnaggi in digital milling? In this regard, I propose to include the following scientific work in the reference section and in the introduction:

1: Pagano S, Moretti M, Marsili R, Ricci A, Barraco G, Cianetti S. Evaluation of  the Accuracy of Four Digital Methods by Linear and Volumetric Analysis of Dental  Impressions. Materials (Basel). 2019 Jun 18;12(12). pii: E1958.

-Line 54 remove "tetragonal"

-Line 80: removing "Crystalline phase, optical, et ...." are useless repetitions

- Figure 2 is not useful for understanding the text

-Line 127 How come the tests were carried out only in a dry environment and not even after storage in water?

-The sample codes are not very clear. Change them, especially in Table 2

-Indicate statistical significance in all tables

-The discussion section should initially provide general indications on the problematic object of the study

-In the final part of the manuscript, the null hypothesis must be expressly rejected

-The conclusions lack a sentence on mechanical properties

-In the references the year of publication must be indicated in bold and all the authors of the references must be indicated (do not insert et al.)

-Figure 1 is unclear: modify it and make it more readable

-Morofolgic analyzes after SEM images are completely missing.These are important for understanding sample variations and making clinical hypotheses

Author Response

I, the author, highly appreciate the detailed valuable comments on this manuscript.

The suggestions are quite helpful for me and I incorporate them in the revised paper.

The revision was listed below the comments and recommendations one by one.

================================================================

-The author's affiliation is missing

I submitted Title page independently which included my affiliation.

-In the abstract a general sentence on the problem should be inserted

I inserted a general sentence on the problem in the abstract of the revised manuscript: Many attempts have been made to improve the translucency of zirconia in dentistry.

-In the abstract section remove each acronym

I mentioned full words first and used the abbreviation next.

-Introduction: some general considerations on the advantages and disadvantages of zirconia compared, for example, to lithium disilicate, should be provided.

Especially considering CAD CAM technologies which best choice for the clinician? And what vatnaggi in digital milling? In this regard, I propose to include the following scientific work in the reference section and in the introduction:

1: Pagano S, Moretti M, Marsili R, Ricci A, Barraco G, Cianetti S. Evaluation of  the Accuracy of Four Digital Methods by Linear and Volumetric Analysis of Dental  Impressions. Materials (Basel). 2019 Jun 18;12(12). pii: E1958.

I mentioned and cited that in the Introduction section of the revised manuscript:

As a result, translucency can be vastly improved with the new cubic phase-containing zirconia, which is comparable to lithium disilicate [6,8].

  1. Pagano, S.; Moretti, M.; Marsili, R.; Ricci, A.; Barraco, G.; Cianetti, S. Evaluation of the accuracy of four digital methods by linear and volumetric analysis of dental impressions. Materials (Basel) 2019, 12, 1958.

-Line 54 remove "tetragonal"

Yes, I did.

-Line 80: removing "Crystalline phase, optical, et ...." are useless repetitions

Yes, I did.

- Figure 2 is not useful for understanding the text

I inserted Figure 2 for better understanding. There are several zirconia-yttria phase diagrams and my study used zirconia-yttria phase diagram presented by Scott in 1975.

-Line 127 How come the tests were carried out only in a dry environment and not even after storage in water?

I tried not to make the dry environment. The bending tests were performed under controlled temperature (22 ± 1°C) and humidity (31 ± 1%).

-The sample codes are not very clear. Change them, especially in Table 2

I made sample codes by using the material’s shade name and used those codes throughout the whole manuscript including Tables and Figures.

I am sorry it was hard to change sample codes.

-Indicate statistical significance in all tables

Tables showed experimental data and I indicated statistical significance in the statistical analysis part in Result section.

-The discussion section should initially provide general indications on the problematic object of the study

I provided general indications on the problematic object of the study in the discussion section of the revised manuscript: In order to further improve the translucency of dental monolithic zirconia, the author fabricated t’-phase-containing translucent zirconia.

-In the final part of the manuscript, the null hypothesis must be expressly rejected

I mentioned in the discussion section: According to the results of this study, a rapid-cooling protocol found to be detrimental to the mechanical properties even though it improved the translucency of dental monolithic zirconia containing 3 to 5 mol% Y2O3. Therefore, the null hypothesis was rejected.

-The conclusions lack a sentence on mechanical properties

I mentioned mechanical properties in the conclusions of the revised manuscript: The t’-phase can improve the translucency, while its mechanical properties were sacrificed. The optimal protocol to improve the translucency without deteriorating its mechanical characteristics of dental monolithic zirconia will be determined in the future.

-In the references the year of publication must be indicated in bold and all the authors of the references must be indicated (do not insert et al.)

I indicated the year in bold and all the authors of the references.

-Figure 1 is unclear: modify it and make it more readable

Yes, I did.

-Morofolgic analyzes after SEM images are completely missing. These are important for understanding sample variations and making clinical hypotheses

I am sorry. I did not understand “Morofolgic”

Maybe: I analyzed grain size and morphology in SEM images.

Reviewer 3 Report

Dear Author,

Your manuscript is really interesting and it is about a current topic in dentistry.

I think that Your manuscript needs to be improved before publication.

First of all, please follow MDPI guidelines. https://www.mdpi.com/authors

In keyword section please use medical subject headings (MeSH). https://meshb.nlm.nih.gov/search

In introduction section, the aim of the study should be clearer, maybe You could subdivide introduction section into two subparagraph (background and aim).

In discussion section You could refer to some recent publication:

Cervino, G.; Fiorillo, L.; Arzukanyan, A.; Spagnuolo, G.; Campagna, P.; Cicciù, M. Application Of Bioengineering Devices For The Stress Evaluation In Dentistry: The Last 10 Years Fem Parametric Analysis Of Outcomes And Current Trends. Minerva Stomatologica 2020.

Cervino, G.; Fiorillo, L.; Arzukanyan, A.V.; Spagnuolo, G.; Cicciu, M. Dental Restorative Digital Workflow: Digital Smile Design from Aesthetic to Function. Dent J (Basel) 2019, 7, doi:10.3390/dj7020030.

Please improve conclusion section and refer to future perspective of the study, do not make a list.

Please insert figures into the text and not at the end. 

Author Response

I, the author, highly appreciate the detailed valuable comments on this manuscript.

The suggestions are quite helpful for me and I incorporate them in the revised paper.

The revision was listed below the comments and recommendations one by one.

================================================================

- In keyword section please use medical subject headings (MeSH). https://meshb.nlm.nih.gov/search

I used medical subject headings in keyword section: Phase transition

- In introduction section, the aim of the study should be clearer, maybe You could subdivide introduction section into two subparagraph (background and aim).

I mentioned the aim of the study in the introduction section of the revised manuscript: Taking advantage of the temperature-dependent phase transformation behavior of zirconia, the author induced t’-phase and investigated the role of t’-phase in dental zirconia in this report. The purposes of this in vitro study were therefore to analyze the crystalline phases of dental monolithic zirconia with 3 to 5 mol% Y2O3 and to verify the effect of a rapid-cooling heat treatment on the crystalline structures, optical, and mechanical properties of dental monolithic zirconia.

- In discussion section You could refer to some recent publication:

Cervino, G.; Fiorillo, L.; Arzukanyan, A.; Spagnuolo, G.; Campagna, P.; Cicciù, M. Application Of Bioengineering Devices For The Stress Evaluation In Dentistry: The Last 10 Years Fem Parametric Analysis Of Outcomes And Current Trends. Minerva Stomatologica 2020.

Cervino, G.; Fiorillo, L.; Arzukanyan, A.V.; Spagnuolo, G.; Cicciu, M. Dental Restorative Digital Workflow: Digital Smile Design from Aesthetic to Function. Dent J (Basel) 2019, 7, doi:10.3390/dj7020030.

 I referred the second one in the revised manuscript: 2. Cervino, G.; Fiorillo, L.; Arzukanyan, A.V.; Spagnuolo, G.; Cicciù, M. Dental restorative digital workflow: digital smile design from aesthetic to function. Dent. J. (Basel) 2019, 7, 30.

- Please improve conclusion section and refer to future perspective of the study, do not make a list.

Please insert figures into the text and not at the end. 

I modified the end part of the discussion section and conclusion section of the revised manuscript: Although there have been much research on the fabrication of high-translucent monolithic zirconia with an addition of dopant element, there have been no research on the production of t’-phase-containing translucent zirconia. Due to the high overlapping diffraction pattern with c-phase reflection, t’-phase can be easily missed. In this study, it was found that sintered/polished dental monolithic zirconia with 3 to 5 mol% Y2O3 contained t’-phase instead of c-phase, as observed by the Rietveld analysis. Particularly, the author tried to produce t’-phase through a rapid cooling protocol according to the temperature-dependent phase transition mechanism of zirconia. Based on the results obtained, the formation of t’-phase increased the translucency of zirconia and t’-phase-containing translucent monolithic zirconia could be fabricated not with an addition of dopant element but with the change of phase composition through a specific cooling protocol.

In summary, the author fabricated t’-phase-containing translucent monolithic zirconia by altering phase compositions using a rapid-cooling protocol. The t’-phase can improve the translucency, while its mechanical properties were sacrificed. The optimal protocol to improve the translucency without deteriorating its mechanical characteristics of dental monolithic zirconia will be determined in the future.

I inserted figures and tables into the revised manuscript.

Reviewer 4 Report

Dear Authors,

The present study deals with the evaluation of the effect of a rapid-cooling heat treatment on the optical and mechanical properties of dental monolithic zirconia. Some changes should be performed in the manuscript in order to improve its quality. First of all, the introduction should be modified in order to better introduce the context of the study. The authors are asked to clearly state the originality and novelty of the study and to clearly explain studies existing in scientific literature, that offer details about the same subject. Comparison with existing data must also highlight the novelty of the proposed study. Some changes in the introduction must also be performed, as it follows:

  • Line 31-32: please remove “to enhance its translucency” to avoid repetition
  • Line 33-34: first sentence repeats the statement above; please change to introduce the explanation of strategies
  • Line 41: please introduce something to connect the paragraph with the previous ideas
  • Line 68-69: please explain sentence “However, there have been few studies that investigated the role of t’-phase in dental zirconia.” and define “few”

Moreover, in the presentation of results and discussion, please include figures and tables and change the manuscript accordingly. At the same time, as at the end of discussions you state the limitations of your study, please add perspectives and re-state however the novelty of the present study. It is not adequate to end a research article this way, you should state the discoveries that your study has brought and which should enrich scientific literature and data on the subject. In the conclusion of the study, please refer to the findings, not to limitations. You need to state the findings of yours study in this section too, not the limitations.

Authors are also asked to perform a language check and an ortograph checkin for spelling mistakes (e.g. in vitro should be italic and before and you cannot put a comma (,)).

Author Response

I, the author, highly appreciate the detailed valuable comments on this manuscript.

The suggestions are quite helpful for me and I incorporate them in the revised paper.

The revision was listed below the comments and recommendations one by one.

================================================================

Line 31-32: please remove “to enhance its translucency” to avoid repetition

Yes, I removed.

Line 33-34: first sentence repeats the statement above; please change to introduce the explanation of strategies

I changed to introduce the explanation of strategies in the revised manuscript: Along with those attempts, the changes in the mechanical and aging behaviors have been addressed.

Line 41: please introduce something to connect the paragraph with the previous ideas

I introduced something to connect the paragraph with the previous ideas: The markedly improved translucency was achieved by using a higher yttria content.

Line 68-69: please explain sentence “However, there have been few studies that investigated the role of t’-phase in dental zirconia.” and define “few” Moreover, in the presentation of results and discussion, please include figures and tables and change the manuscript accordingly. At the same time, as at the end of discussions you state the limitations of your study, please add perspectives and re-state however the novelty of the present study. It is not adequate to end a research article this way, you should state the discoveries that your study has brought and which should enrich scientific literature and data on the subject. In the conclusion of the study, please refer to the findings, not to limitations. You need to state the findings of yours study in this section too, not the limitations.

I removed and explained in the discussion section and I changed discussion section and conclusion sections: Although there have been much research on the fabrication of high-translucent monolithic zirconia with an addition of dopant element, there have been no research on the production of t’-phase-containing translucent zirconia. Due to the high overlapping diffraction pattern with c-phase reflection, t’-phase can be easily missed. In this study, it was found that sintered/polished dental monolithic zirconia with 3 to 5 mol% Y2O3 contained t’-phase instead of c-phase, as observed by the Rietveld analysis. Particularly, the author tried to produce t’-phase through a rapid cooling protocol according to the temperature-dependent phase transition mechanism of zirconia. Based on the results obtained, the formation of t’-phase increased the translucency of zirconia and t’-phase-containing translucent monolithic zirconia could be fabricated not with an addition of dopant element but with the change of phase composition through a specific cooling protocol.

In summary, the author fabricated t’-phase-containing translucent monolithic zirconia by altering phase compositions using a rapid-cooling protocol. The t’-phase can improve the translucency, while its mechanical properties were sacrificed. The optimal protocol to improve the translucency without deteriorating its mechanical characteristics of dental monolithic zirconia will be determined in the future.

I include figures and tables in the revised manuscript.

Authors are also asked to perform a language check and an ortograph checkin for spelling mistakes (e.g. in vitro should be italic and before and you cannot put a comma (,)).

Yes, I did.

Reviewer 5 Report

Dear Author,

The reviewed manuscript describes the procedure and results of experimental studies regarding the effect of heat treatment on the properties of sintered zirconia intended for dental material. While reading the submission, I noticed several issues that could improve the article:

  1. Format the entire article according to the journals guidelines: transfer figures to the text.
  2. 18: change: "Mpa" is "MPa".
  3. 94: "a few minutes" - it's too general.
  4. 109: please explain the symbols in the equation.
  5. 121 and further: add spaces before units
  6. 152: "change:" significant "is" significance ".
  7. chapter 3.2: write zeros before the dot in numbers.
  8. 195: There is no information about the grain size for 4Y.
  9. 209: error in the citation record [4].
  10. References should be formatted and completed. Add doi. Currently, most items are older than 5 years. I suggest citing current (last 3 years) positions from Mdpi publishing house. This will definitely positively affect the visibility of the article.

Author Response

I, the author, highly appreciate the detailed valuable comments on this manuscript.

The suggestions are quite helpful for me and I incorporate them in the revised paper.

The revision was listed below the comments and recommendations one by one.

================================================================

Format the entire article according to the journals guidelines: transfer figures to the text.

I transferred Figures and Tables to the text.

18: change: "Mpa" is "MPa".

Yes, I changed.

94: "a few minutes" - it's too general.

I changes it in the revised manuscript: they were rapidly air-cooled within 1-2 minutes

109: please explain the symbols in the equation.

I explained symbols in the equation in the revised manuscript: ,where and are the differences in chroma and hue for a pair of samples. , , and are the weighting functions for the lightness, chroma, and hue and the parametric factors, , , and are the correction terms for variations in experimental conditions. , a rotation function, accounts for the interaction between chroma and hue differences in the blue region.

121 and further: add spaces before units

Yes. I did.

152: "change:" significant "is" significance ".

The significance level was set at 0.05.

chapter 3.2: write zeros before the dot in numbers.

I wrote zeros before the dot in numbers in Result section of the revised manuscript.

195: There is no information about the grain size for 4Y.

I mentioned the grain size for 4Y in the revised manuscript: SEM images of each ceramic group were shown in Figure 5. The grain sizes of sintered/polished 3Y-TZP ranged from about 280 to 350 nm, whereas larger grains were observed in 4Y- (416-890 nm) or 5Y-PSZ (543-1060 nm).

209: error in the citation record [4].

I changed. Thank you.

References should be formatted and completed. Add doi. Currently, most items are older than 5 years. I suggest citing current (last 3 years) positions from Mdpi publishing house. This will definitely positively affect the visibility of the article.

I added doi in the references.

I cited current positions from Mdpi publishing house: Reference 2, 8.

Round 2

Reviewer 1 Report

The second version of the paper is a poor improvement, and a substantial revision is needed to make this manuscript suitable for publication.

  1. The author sintered the S2, E2, and SMS2 samples at 1500º But the R/S2, R/E2, and R/SMS2 samples were sintered at 1550ºC and then rapidly cooled. Is this right? The final sintering temperature and total sintering time induce the larger crystal size and high translucency. Why did the author change the sintering temperature?
  2. The author should show the p values of TP and T% between each sintered/polish and rapid cooling group. Remove one bracket of T% SD of R/SMS2.
  3. Thank you for indicating the R factors. These data are also fruitful for the audience and would be shown in the manuscript.
  4. The auhots should show the figure numbers of Figure 3.
  5. What is the Figure 3(g)? Optical photograph or color chart? The T% value of Figure 3(h) seems to be different from the data in Table 2.
  6. The resolution of Figures 4 and 5 still unclear. Did the author check the draft in PDF files before submission?
  7. Please show the correct figure numbers of Figure 4. (b) and (d) should be incorrect.

Author Response

I, the author, highly appreciate the detailed valuable comments on this manuscript.

The suggestions are quite helpful for me and I incorporate them in the revised paper.

The revision was listed below the comments and recommendations one by one.

================================================================

  1. The author sintered the S2, E2, and SMS2 samples at 1500º But the R/S2, R/E2, and R/SMS2 samples were sintered at 1550ºC and then rapidly cooled. Is this right? The final sintering temperature and total sintering time induce the larger crystal size and high translucency. Why did the author change the sintering temperature?

I sintered the S2, E2, and SMS2 samples at 1500º, but the R/S2, R/E2, and R/SMS2 samples were sintered at 1550ºC and then rapidly cooled. I changed the sintering temperature in order to being employed the samples inside the two-phase (c + t) field, taking into account Scott’s phase diagram.

  1. The author should show the p values of TP and T% between each sintered/polish and rapid cooling group. Remove one bracket of T% SD of R/SMS2.

I showed the p values in Table 2 and removed one bracket of T% SD of R/SMS2. Thank you.

  1. Thank you for indicating the R factors. These data are also fruitful for the audience and would be shown in the manuscript.

I showed R factors in Table 4.

  1. The authors should show the figure numbers of Figure 3.

I showed figure numbers of Figure 3.

  1. What is the Figure 3(g)? Optical photograph or color chart? The T% value of Figure 3(h) seems to be different from the data in Table 2.

Figure 3(g) is color chart data from the spectrophotometer.

Figure 3(h) is one of the E2 and R/E2 samples data.

  1. The resolution of Figures 4 and 5 still unclear. Did the author check the draft in PDF files before submission?

I changed Figures 4 and 5 with high-resolution data for the revised manuscript.

  1. Please show the correct figure numbers of Figure 4. (b) and (d) should be incorrect.

I showed the correct figure numbers of Figure 4(b) and (d).

Thank you for your advice.

Reviewer 2 Report

Some questiones were accept ed. I suggest manuscript acceptation 

Author Response

I, the author, highly appreciate the detailed valuable comments on this manuscript.

Thank you.

Reviewer 4 Report

Dear Authors,

The article is significantly improved after the changes you performed, as suggested. Nevertheless, please take care of your bibliography and arrange it accordingly to the journal demands.

Author Response

I, the author, highly appreciate the detailed valuable comments on this manuscript.

The suggestions are quite helpful for me and I incorporate them in the revised paper.

The revision was listed below the comments and recommendations one by one.

================================================================

The article is significantly improved after the changes you performed, as suggested. Nevertheless, please take care of your bibliography and arrange it accordingly to the journal demands.

Thank you. I re-arranged the reference No. 18, 21, 22 according to the journal demands.

Round 3

Reviewer 1 Report

Please revise line 122 and 129. Figure size should be modified.